# Systemic and local cues drive neural stem cell niche remodelling during neurogenesis in *Drosophila*

**Pauline Spéder[1,2†], Andrea H Brand[1,2*]**

[1]Department of Physiology, Development and Neuroscience, University of Cambridge, Cambridge, United Kingdom; [2]The Gurdon Institute, University of Cambridge, Cambridge, United Kingdom

**Abstract** Successful neurogenesis requires adequate proliferation of neural stem cells (NSCs) and their progeny, followed by neuronal differentiation, maturation and survival. NSCs inhabit a complex cellular microenvironment, the niche, which influences their behaviour. To ensure sustained neurogenesis, niche cells must respond to extrinsic, environmental changes whilst fulfilling the intrinsic requirements of the neurogenic program and adapting their roles accordingly. However, very little is known about how different niche cells adjust their properties to such inputs. Here, we show that nutritional and NSC-derived signals induce the remodelling of *Drosophila* cortex glia, adapting this glial niche to the evolving needs of NSCs. First, nutrition-induced activation of PI3K/Akt drives the cortex glia to expand their membrane processes. Second, when NSCs emerge from quiescence to resume proliferation, they signal to glia to promote membrane remodelling and the formation of a bespoke structure around each NSC lineage. The remodelled glial niche is essential for newborn neuron survival.

DOI: https://doi.org/10.7554/eLife.30413.001

**\*For correspondence:**
a.brand@gurdon.cam.ac.uk

**Present address:** [†]Department of Developmental and Stem Cell Biology, InstitutPasteur, CNRS UMR3738, Paris, France

**Competing interests:** The authors declare that no competing interests exist.

## Introduction

Stem cell niches support the normal function of stem cells (*Bjornsson et al., 2015*; *Lander et al., 2012*). The mammalian NSC niche displays an intricate and compact architecture made up of diverse cell populations, including neurons, astrocytes, blood vessels forming part of the blood-brain barrier (BBB) and resident immune cells, the microglia (*Bjornsson et al., 2015*; *Silva-Vargas et al., 2013*). Both mechanical and diffusible signals pass between cell populations to influence NSCs (*Bjornsson et al., 2015*; *Silva-Vargas et al., 2013*). Local astrocytes are in close association with both NSCs and the vasculature. In addition, stem cells contact endothelial cells, receiving systemic inputs from the blood. Combinatorial and coordinated interactions between different niche cells are thought to provide the permissive environment for appropriate neurogenesis (*Goldman and Chen, 2011*). How niche cells respond to NSC inputs and how these interactions respond to external pressures remain to be determined.

To investigate how niche cells adapt to systemic and local changes to support neurogenesis, we turned to a simpler model, the post-embryonic central nervous system (CNS) of *Drosophila melanogaster*. The *Drosophila* larval CNS comprises the central brain (CB), the optic lobe (OL) and the ventral nerve cord (VNC). Larval NSCs are found in a microenvironment that includes neurons, a blood-brain barrier and a variety of different glia that structure the brain in layers (*Figure 1—figure supplement 1A*) (*Ito et al., 1995*; *Stork et al., 2012*). Several of these populations modulate NSC behaviour, thus acting as niche cells. *Drosophila* NSCs go through two distinct rounds of neurogenesis (*Figure 1A*) (*Truman and Bate, 1988*; *Egger et al., 2008*; *Kang and Reichert, 2015*). They proliferate actively during embryogenesis to generate primary neurons that form the functional larval

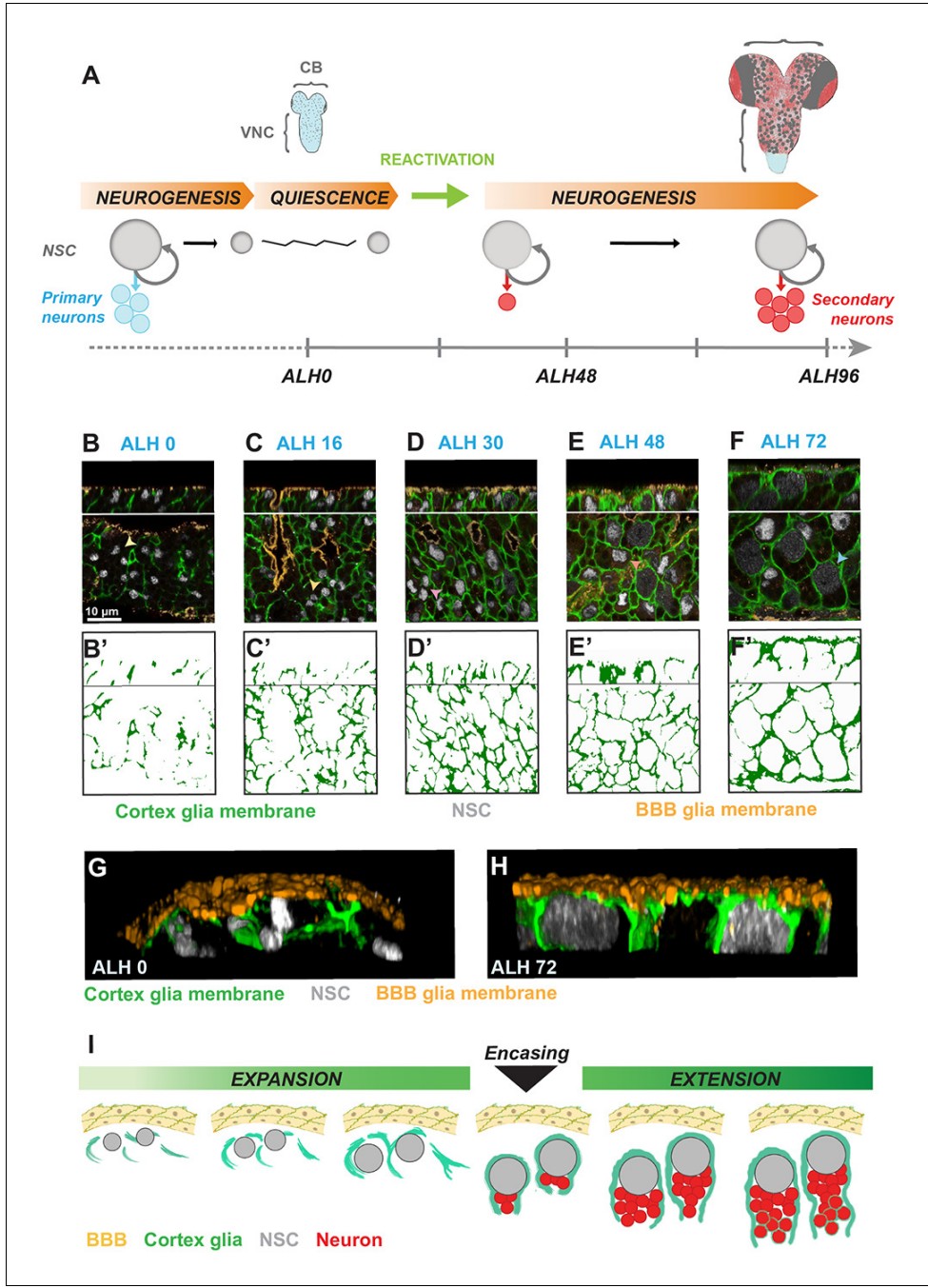

**Figure 1.** Individual *Drosophila* neural stem cells and their lineages are progressively enclosed in cortex glial chambers. (**A**) Two waves of neurogenesis take place in the *Drosophila* CNS. NSCs divide asymmetrically during embryogenesis to form the larval nervous system. They then enter quiescence, from which they reactivate at early larval stages to generate neurons that build most of the adult nervous system. CB, central brain. VNC, ventral nerve cord. (**B–F'**) Progressive formation of cortex glial chambers. Five time points were assessed : ALH0, ALH16, ALH30, ALH48 and ALH72 (at 25°C). (**B–F**) Top and bottom panels, orthogonal and ventral views of a fragment of the VNC. (**B'–F'**) Cortex glial membrane signal outlines. (**G–H**) 3D reconstruction of an orthogonal view of a fragment of one VNC. (**G**) After larval hatching and H) at a late larval stage. Genotype : *Nrv2::GFP, moody-GAL4 ; UAS-mCD8-RFP*. (**I**) Sketch representing the timeline of cortex glial chamber formation. BBB, RFP, orange ; cortex glial membrane, Nrv2::GFP, green ; NSC, Deadpan, grey.

DOI: https://doi.org/10.7554/eLife.30413.002

The following figure supplement is available for figure 1:

*Figure 1 continued on next page*

*Figure 1 continued*

**Figure supplement 1.** Organisation and formation of the cortex glial chamber.

DOI: https://doi.org/10.7554/eLife.30413.003

nervous system. NSCs then become mitotically dormant, entering a quiescent phase. Post embryonically, in response to nutrition, NSCs awaken almost synchronously (*Britton and Edgar, 1998*) and enter a second neurogenic program. The reactivation and proliferation of these central brain and ventral nerve cord NSCs ultimately generate secondary neuronal lineages that form the adult nervous system.

The *Drosophila* BBB acts as a signalling interface between the hemolymph (the *Drosophila* equivalent of blood) and brain cells (*Stork et al., 2008*; *DeSalvo et al., 2011*). It is exclusively of glial nature, formed by a layer of perineurial glia and a layer of subperineurial glia, while the vertebrate BBB is a composite of endothelial cells and glial cells (astrocytes) (*Stork et al., 2008*). Both fulfill neuroprotective roles, relying on a physical paracellular barrier (tight junctions of the vertebrate endothelial cells and septate junctions of the *Drosophila* subperineurial glia). Importantly, the *Drosophila* BBB mediates the impact of nutrition on NSC reactivation (*Chell and Brand, 2010*; *Spéder and Brand, 2014*). Essential amino acids in the larval diet trigger the local production and secretion of insulin-like peptides (dILPs) by the subperineurial glial layer of the BBB (*Chell and Brand, 2010*; *Spéder and Brand, 2014*). dILPs bind to the Insulin/IGF receptor on NSCs, activating the conserved PI3K/Akt pathway (*Chell and Brand, 2010*; *Sousa-Nunes et al., 2011*). Consequently, NSCs enlarge and re-enter the cell cycle (*Figure 1—figure supplement 1B*). These reactivated, actively cycling NSCs are found in close association with the cortex glia (*Hoyle, 1986*; *Hoyle et al., 1986*; *Pereanu et al., 2005*). This association is known to protect NSCs from oxidative stress and nutritional restriction during late larval stages (*Cheng et al., 2011*; *Bailey et al., 2015*).

Cortex glia display a remarkable organisation around NSCs and neurons (*Pereanu et al., 2005*). Each NSC and its progeny is individually enwrapped in a chamber formed of cortex glial membrane, which separates the lineages from one another (*Figure 1—figure supplement 1A and C–D*"). The lineage is organised within the chamber, with the NSC at the top and the newly-born neurons being eventually pushed down (*Figure 1—figure supplement 1E*). Notably, younger neurons are enclosed in the same chamber as their mother NSC, whereas older neurons display their own individual chambers (*Pereanu et al., 2005*; *Dumstrei et al., 2003*). Primary neurons are also surrounded by cortex glia membrane (*Pereanu et al., 2005*).

## Results

An important question has been whether the cortex glial chamber is present throughout postembryonic life, enclosing NSC lineages from quiescence to proliferation (*Sousa-Nunes et al., 2011*; *Limmer and Klämbt, 2014*), or whether the cortex glial niche evolves over time (*Pereanu et al., 2005*). To address this, we used a protein trap that labels cortex glial membranes (Nrv2::GFP) to follow the association between cortex glia and NSCs throughout larval life (*Figure 1B–F'*). At ALH0 (ALH, hours after larval hatching), quiescent NSCs were not separated from each other by cortex glial membranes. Cortex glial membranes then progressively underwent three developmental steps. First, membranes expanded. This expansion phase lasted for about a third of larval life. At this stage, NSCs were still not enclosed individually (*Figure 1C–D'*). Second, membranes fully encased each NSC lineage, forming a chequerboard of cortex glia chambers (*Figure 1E–E'*). We call this event "encasing". Finally, during an extension phase, cortex glial adapted to lineage expansion while maintaining chamber organisation (*Figure 1F–F'* and data not shown for ALH96). The use of another cortex glial membrane marker confirmed these results (*Figure 1—figure supplement 1F–G*).

To gain further insight into the relationship between cortex glia, NSCs and the BBB, we analysed 3D reconstructions. Interestingly, at ALH0 NSCs were not covered by a sheet of cortex glial membrane, but directly abutted the BBB layer (*Figure 1G*). However, at ALH72 the cortex glial membrane fully encased NSCs, isolating NSCs from the BBB (*Figure 1H*). Importantly, the absence of cortex glial chambers at early stages enables BBB-secreted insulin-like peptides to reach NSCs, consequently triggering NSC reactivation. However, at late larval stages, when NSC reactivation is complete (*Chell and Brand, 2010*; *Sousa-Nunes et al., 2011*) and proliferation no longer depends on

insulin signalling (*Cheng et al., 2011*), the cortex glial membranes generate a dense interface. These results indicate that cortex glial chambers evolve throughout larval life. Furthermore, the evolution of cortex glial membrane morphology strikingly parallels NSC reactivation, including an initial growth phase followed by a decisive switch: chamber formation on the one hand and mitotic entry on the other (*Figure 1I*).

We assessed whether cortex glial chamber formation was also controlled by nutrition, the key trigger of NSC reactivation. We first asked whether removing the nutritional signal was sufficient to block cortex glial chamber formation. Unlike control fed larvae, larvae starved for 72 hr did not develop a sealed chequerboard structure around NSCs, (*Figure 2A–B'*). To better describe and quantify chamber organisation defects, we plotted normalised intensities of cortex glial membrane, NSCs and neuron signals against depth (see Materials and methods for details). Wild-type cortex glial chambers, from fed larvae, show a characteristic graph profile (*Figure 2C*). The cortex glial signal peaks first, as a readout of the dense membrane layer forming on top of NSCs, just under the BBB. Next the peak corresponding to NSCs appears, situated in the upper part of the chamber. Finally, the last peak corresponds to the newborn neurons under the NSCs. In contrast, in starved larvae cortex glia, NCSc and neurons all peak at the same depth, pinpointing a lack of chamber order (*Figure 2D–E*). We also calculated the ratio of cortex glial membrane to NSC number in both conditions. Starvation resulted in a loss of cortex glial membrane (*Figure 2F*). Altogether these results demonstrate that nutrition is critical for cortex glial chamber formation.

We tested whether nutrition acts on cortex glial chamber formation through insulin-like peptide secretion from the BBB. Insulin-like peptide secretion relies on gap junction-dependent coordinated calcium oscillations in the BBB (*Spéder and Brand, 2014*). First, we performed a gap junction knock-down in the BBB. This resulted in strong chamber defects (*Figure 2—figure supplement 1A–A'*), suggesting that BBB-secreted insulin-like peptides are indeed required for cortex glial chamber formation. To identify which dILPs are expressed by the BBB at the time of membrane growth, we determined RNA Pol II binding in the subperineurial glia of fed larvae (Jessie Van Buggenum, P.S. and A.H.B., unpublished). We found Pol II binding only at *dilp6* (*Figure 2—figure supplement 1B*). Second, we asked if nutrition activates the insulin/PI3K/Akt pathway specifically in cortex glial cells. We identified a new driver, *cyp4g15-GAL4*, whose expression is highly restricted to cortex glial cells (*Figure 2—figure supplement 1C–I*). Impairing insulin signalling only in the cortex glia, by expressing Δp60 which has a dominant-negative effect on PI3K activity, resulted in incomplete NSC encasing (*Figure 3A–B'*). More precisely, formation of the top cortex membrane layer was incomplete and erratic, with NSCs touching one another (see arrowheads). Cellular organisation of the chamber was lost, as shown by the disordered plots and peak values between cortex glia, NSCs and neurons (*Figure 3C–E*). Furthermore, the ratio of cortex glial membranes to NSC number strongly decreased (*Figure 3F*). Notably, affecting insulin signalling directly at the level of the insulin receptor, by expressing a dominant negative form (2X InR$^{DN}$), also led to impaired chamber formation, with loss of cellular organisation and strongly decreased membrane signal (*Figure 3—figure supplement 1*). These data demonstrate that direct integration of dILP6 binding and insulin signalling are required in the cortex glia for NSC chamber building. In addition, they show that chamber loss results in NSC lineage disorganisation, as was suggested previously (*Pereanu et al., 2005*). Thus, formation of the cortex glial chamber relies on an external systemic stimulus, nutrition.

Next we assessed whether activating insulin signalling in the cortex glia was sufficient to rescue chamber formation in starved larvae (*Figure 3G–H'*). Starvation maintains NSCs in a quiescent state. Expression of a constitutively active form of Akt (Akt$^{Act}$) specifically in the cortex glia neither rescued individual NSC enclosing, nor chamber organisation (*Figure 3I–K*). However, it led to an increase in cortex membrane ratio to NSC number, compared to the starved control (*Figure 3L*), indicating that insulin signalling autonomously promotes membrane expansion. Surprisingly, cortex glial membranes still wrapped around primary neurons, providing them with tailored chambers (*Figure 3H–H'*). This could explain the increase in cortex glial signal above NSCs (*Figure 3K*, column 2). These results indicate that PI3K/Akt pathway activation in the cortex glia alone is not sufficient to re-establish chamber formation under starvation.

Interestingly, cortex glia start to divide actively around the time chambers form (ALH48, see (*Avet-Rochex et al., 2012*) and our own observation). Moreover, cortex glial division is partially under the control of the insulin receptor (*Avet-Rochex et al., 2012*). Therefore, we checked that cortex glial division did not account for insulin-dependent chamber formation. Indeed, hampering

cortex glial proliferation through knockdown of *string* (*stg*, the *Drosophila cdc25*) did not prevent NSC chamber formation (*Figure 3—figure supplement 2*). Similar results were obtained by expressing *dacapo*, an inhibitor of cell cycle progression (*Figure 3—figure supplement 2*). These results

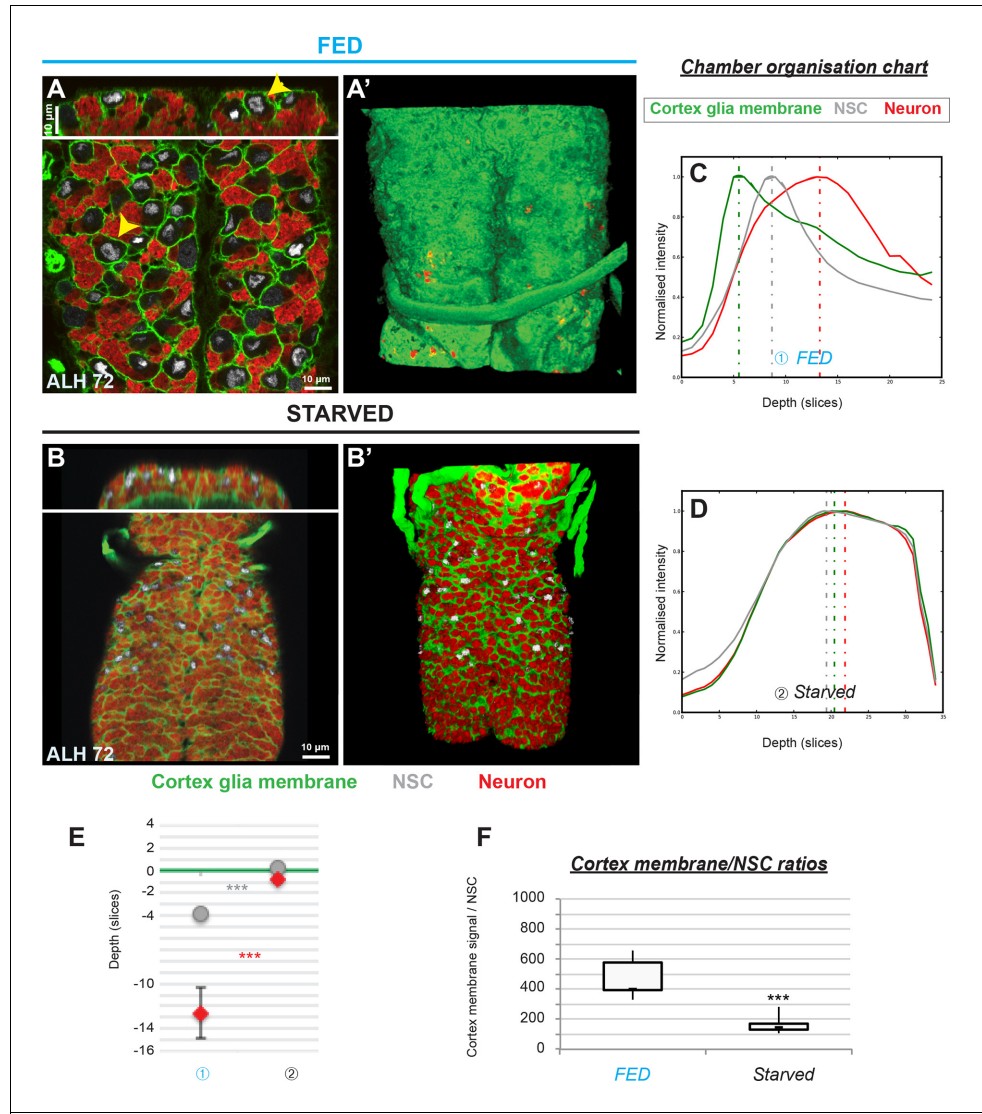

**Figure 2.** Cortex glial remodelling depends on nutritional cues. (A–A') Cortex glia enclose each NSC and its lineage into an individual membrane chamber in control fed larvae. (A) Bottom and top panels, ventral and orthogonal views of a third instar VNC. Yellow arrowheads show examples of individual chambers. (A') 3D reconstruction, ventral view. (B–B') Cortex glial chambers are not formed under starvation. (B) Top and bottom panels, orthogonal and ventral views of a fragment of one VNC. (B') 3D reconstruction, ventral view. (C–D) Chamber organisation charts for (C) one control fed and (D) one starved larva. The graph represents signal intensities from cortex glial membranes (green), NSCs (grey) and neurons (red) plotted against depth for one VNC. Depth unit is the slice. (E) Statistical representation of NSC (grey) and neuron (red) Z positions relative to the top layer of the cortex glia (green line). p(NSC)=$1.4*10^{-3}$, p(neuron)=$5.2*10^{-4}$. n (FED) = 6 VNCs and n (Starved) = 6 VNCs. (F) Ratios between cortex glial membrane signal and NSCs. p=$5.0*10^{-4}$. n (Fed) = 6 VNCs. n (Starved) = 6 VNCs. Cortex glial membrane, Nrv2::GFP, green ; NSC, Deadpan, grey ; neurons, ElaV, red.

DOI: https://doi.org/10.7554/eLife.30413.004

The following figure supplement is available for figure 2:

**Figure supplement 1.** NSC chamber formation is nutrition-dependent.
DOI: https://doi.org/10.7554/eLife.30413.005

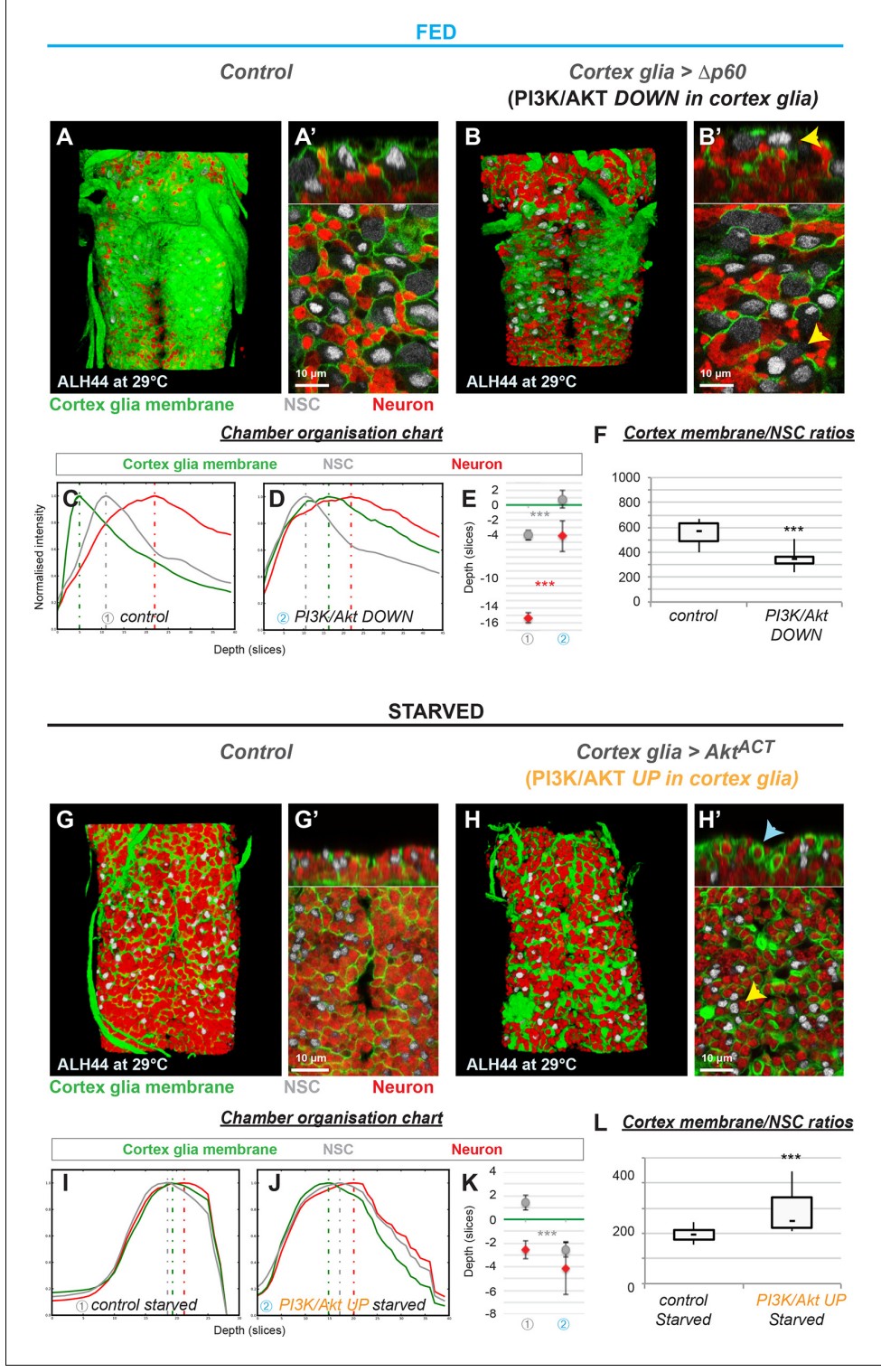

**Figure 3.** The insulin pathway is required in the cortex glia for autonomous chamber growth. (**A–F**) Knockdown of insulin signalling specifically in the cortex glia impairs chamber formation. (**A**) 3D reconstruction of the thoracic section of one fed control VNC, ventral view. (**A'**) Top and bottom panels, orthogonal and ventral views. (**B**) 3D reconstruction of the thoracic section of one VNC of Δp60 overexpression at ALH60, ventral view. (**B'**) Top and bottom panels, orthogonal and ventral views. Yellow arrowheads show examples of uncased NSCs. (**C–E**) Chamber organisation chart. The graph represents signal intensities from cortex glial membranes (green), NSCs (grey) and neurons (red) plotted against depth for one VNC in (**C**) control and (**D**) Δp60 overexpression. Depth unit

*Figure 3 continued on next page*

*Figure 3 continued*

is the slice. (**E**) Statistical representation of NSC (grey) and neuron (red) Z positions relative to the top layer of the cortex glia (green line). p(NSC)=6.5*10$^{-3}$, p(neuron)=6.9*10$^{-4}$. (**F**) Ratios between cortex glial membrane signal and NSCs. p=1.1*10$^{-3}$. n (control) = 6 VNCs. n ($\Delta p60$)=7 VNCs. (**G–L**) Activation of insulin signalling in the cortex glia only is not sufficient for NSC encasing in starved conditions. (**G**) 3D reconstruction of the thoracic section of one starved control VNC, ventral view. (**G'**) Top and bottom panels, orthogonal and ventral views. (**H**) 3D reconstruction of the thoracic section of one VNC of starved *Akt*$^{ACT}$ overexpression, ventral view. (**H'**) Top and bottom panels, orthogonal and ventral views. Yellow arrowheads show examples of unenclosed NSCs. Blue arrowheads show examples of encased primary neurons. (**I–K**) Chamber organisation chart. The graph represents signal intensities from cortex glial membranes (green), NSCs (grey) and neurons (red) plotted against depth for one VNC in I) starved control and (**J**) starved *Akt*$^{ACT}$ overexpression. Depth unit is the slice. (**K**) Statistical representation of NSC (grey) and neuron (red) Z positions relative to the top layer of the cortex glia (green line). p (NSC)=4.5*10$^{-4}$, p(neuron)=0.52. (**L**) Ratios between cortex glial membrane signal and NSCs. p=7.3*10$^{-3}$. n (control) = 9 VNCs. n (Akt$^{ACT}$) = 10 VNCs. For all the stainings : Cortex glial membrane, Nrv2::GFP, green ; NSC, Dpn, grey ; neurons, ElaV, red.

DOI: https://doi.org/10.7554/eLife.30413.006

The following figure supplements are available for figure 3:

**Figure supplement 1.** The insulin receptor is required in the cortex glia for chamber formation.
DOI: https://doi.org/10.7554/eLife.30413.007

**Figure supplement 2.** Cortex glial division is not required for NSC chamber formation.
DOI: https://doi.org/10.7554/eLife.30413.008

---

indicate that cortex glia have a certain plasticity, compensating for population loss to meet NSC demand.

Our data suggest that another signal is necessary to shift cortex glial membrane expansion towards chamber formation. Notably, NSC reactivation is triggered by nutrition and is fully complete around chamber closing time (ALH48, see **Figure 1A and I**). We thus decided to test whether NSC reactivation promotes chamber formation. First, we delayed NSC reactivation by blocking insulin signalling specifically in NSCs using the PI3K inhibitor, PTEN. Although phenotype expressivity was partial, we found that NSC chamber formation was strongly affected in parts of the VNC (**Figure 4A–B'**). Several NSCs were grouped together rather than individually enclosed and cortex glial chamber organisation was altered compared to controls (**Figure 4C–E**). Membrane growth was not significantly decreased (**Figure 4F**). This could be due to the autonomous effect of insulin signalling on cortex glial expansion or to the fact that reactivation was only delayed and not fully blocked. Similar results were obtained with overexpression of either Δp60 or 2X InR$^{DN}$ (**Figure 4—figure supplement 1A–B'**). To confirm the requirement for NSC reactivation in cortex glial chamber organisation, we turned to intrinsic regulators of this process. Spindle matrix proteins have recently been shown to be required for the mitotic re-entry of quiescent NSCs (**Li et al., 2017**). Accordingly, RNAi knockdown of *east* in NSCs resulted in a dramatic loss of the mitotic marker phosphohistone H3 (PH3, **Figure 4—figure supplement 1C**). We found that this condition also led to impaired cortex glial chamber organisation (**Figure 4—figure supplement 1D–H**), whereas membrane growth was little affected (**Figure 4—figure supplement 1I**). These results show that NSC reactivation is crucial to chamber organisation, but is not the main driving force for cortex glial membrane growth under normal, fed conditions.

Next, we forced NSC exit from quiescence by activating insulin/PI3K/Akt signalling in starved NSCs. Strikingly, overexpression of Akt$^{Act}$ resulted in the formation of several individual NSC chambers throughout the VNC (**Figure 4G–H'**, see yellow arrows). Cellular organisation was restored and cortex glial membrane to NSC ratio increased significantly compared to starved control (**Figure 4I–L**). Thus, NSC reactivation is necessary and sufficient to restore chamber formation under starvation. These data show that PI3K/Akt signalling in NSCs is able to promote cortex glial membrane expansion and NSC encasing. In addition, PI3K/Akt activation in cortex glia leads to autonomous membrane growth (**Figure 5A**).

Cortex glial remodelling, NSC cycle re-entry and the generation of progeny appear tightly coordinated. This led us to reflect on the importance of the timely formation of the cortex glial chamber for NSC lineages. We thus assessed NSC and neuronal behaviour when chamber formation is impaired. First, we investigated the effect of chamber formation on NSC survival and proliferation,

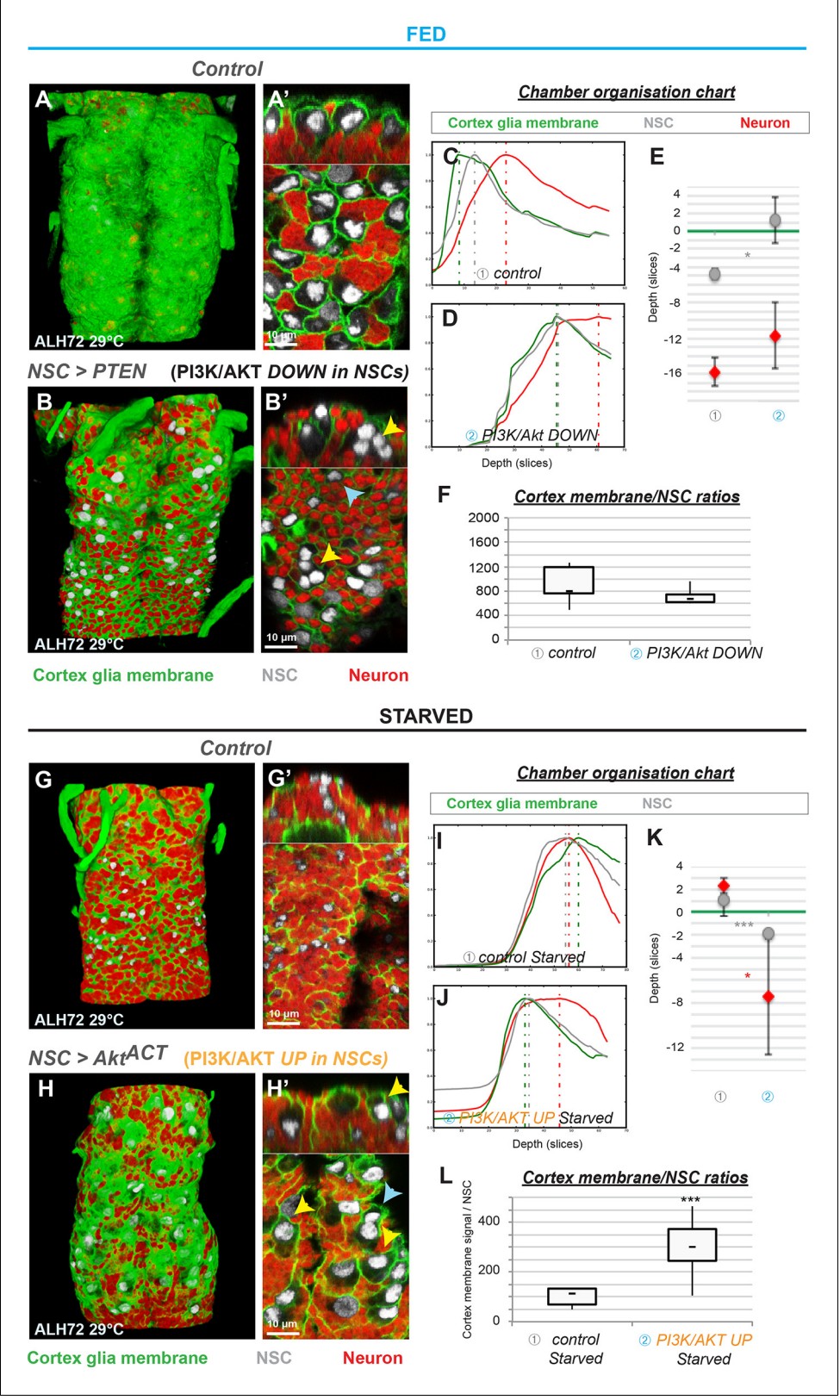

**Figure 4.** The insulin pathway is required in NSCs for non-autonomous cortex glial chamber remodelling. (**A–F**) Knockdown of insulin signalling specifically in NSCs impairs chamber formation. (**A**) 3D reconstruction of the thoracic section of one fed control VNC, ventral view. (**A'**) Top and bottom panels, orthogonal and ventral views. *Figure 4 continued on next page*

*Figure 4 continued*

(B) 3D reconstruction of the thoracic section of one VNC of PTEN overexpression, ventral view. (B') Top and bottom panels, orthogonal and ventral views. Yellow arrowheads show examples of unenclosed NSCs. (C–E) Chamber organisation chart. The graph represents the signal intensities from cortex glial membranes (green), NSCs (grey) and neurons (red) plotted against depth for one VNC in C) control and D) PTEN overexpression. Depth unit is the slice. (E) Statistical representation of NSC (grey) and neuron (red) Z positions relative to the top layer of the cortex glia (green line). p(NSC)=6.6*10$^{-2}$, p(neuron)=0.33. (F) Ratios between cortex glial membrane signal and NSCs. p=0.21. n (control) = 5 VNCs. n (PTEN) = 6 VNCs. (G–L) Activation of insulin signalling in NSCs only is sufficient for chamber remodelling in starved conditions. (G) 3D reconstruction of the thoracic section of one VNC from a starved control condition, ventral view. (G') Top and bottom panels, orthogonal and ventral views. (H) 3D reconstruction of the thoracic section of one VNC from a starved Akt$^{ACT}$ overexpression, ventral view. (H') Top and bottom panels, orthogonal and ventral views. Yellow arrowheads show examples of encased NSCs. Blue arrowheads show examples of unenclosed NSCs. (I–K) Chamber organisation chart. The graph represents the signal intensities from cortex glial membranes (green), NSCs (grey) and neurons (red) plotted against depth for one VNC for I) starved control and J) starved Akt$^{ACT}$ overexpression. Depth unit is the slice. (K) Statistical representation of NSC (grey) and neuron (red) Z positions relative to the top layer of the cortex glia (green line). p (NSC)=4.8*10$^{-2}$, p(neuron)=2.0*10$^{-2}$. (L) Ratios between cortex glial membrane signal and NSCs. p=7.8*10$^{-3}$. n (control) = 5 VNCs. n (Akt$^{ACT}$) = 6 VNCs. For all the stainings : Cortex glial membrane, Nrv2::GFP, green ; NSC, Dpn, grey ; neurons, ElaV, red.

DOI: https://doi.org/10.7554/eLife.30413.009

The following figure supplement is available for figure 4:

**Figure supplement 1.** NSC reactivation is crucial for cortex glial chamber organisation.

DOI: https://doi.org/10.7554/eLife.30413.010

knocking down the PI3K/Akt pathway in cortex glia (see *Figures 3* and *5A*). NSC numbers were unaffected, showing that NSC survival is not compromised when the chamber cannot be built (*Figure 5—figure supplement 1A*). We then determined the cell cycle profile of NSCs by staining for phosphohistone H3 (PH3) and assessing EdU incorporation. We observed a small, but significant increase in PH3$^{+}$ and PH3$^{-}$ EdU$^{-}$ NSCs, at the expense of EdU$^{+}$ NSCs (*Figure 5—figure supplement 1B*).

We then turned to NSC progeny, wondering if a primary function of cortex glia during late larval stages is to support newborn neurons. Using TUNEL staining, we determined the degree of neuronal cell death when chamber formation is impaired. Whereas control conditions showed few TUNEL$^{+}$ neurons (*Figure 5B*), PI3K/Akt pathway knockdown in cortex glia induced pronounced TUNEL$^{+}$ neuron staining (*Figure 5C–D*). In addition, VNC total neuronal volume decreased in this condition compared to controls (*Figure 5—figure supplement 1C*), suggesting that neuron number is decreased when chamber formation is impaired. We thus propose that preventing cortex glial chamber formation leads to increased neuronal apoptosis. In contrast, a condition where cortex glial behaviour is affected without chamber impairment (*stg* knockdown in cortex glia, see *Figure 3—figure supplement 2*) did not alter NSC division or the number of TUNEL$^{+}$ neurons (*Figure 5—figure supplement 1D–G*). Altogether these results indicate that preventing cortex glial chamber formation has little impact on NSC survival and proliferation, but is crucial for newborn neuron survival.

## Discussion

Here, we demonstrate that the cortex glial niche cells are able to integrate both external, nutritional signals and local NSC behaviour, triggering remodelling of their architecture. The encasing of each NSC lineage is in turn essential for successful neurogenesis by ensuring the survival of newborn neurons (*Figure 5E*). Importantly, our work reveals a previously unidentified NSC lineage to cortex glial signal. NSC reactivation, sufficient for triggering lineage encasing, leads to the production of new neurons. NSC lineages are enclosed at about the time of the first NSC division, before any substantial production of neurons. Although we cannot completely rule out that neurons contribute to lineage encasing, we propose that NSCs are the signal-sending cells due to the timing of the event.

The instructive signals sent by NSCs to remodel the cortex glia could be either mechanical or secreted. Interestingly, blocking endocytosis and vesicle recycling in NSCs through dynamin inhibition (*Figure 5—figure supplement 2*) has a limited impact on chamber formation, mostly leading to

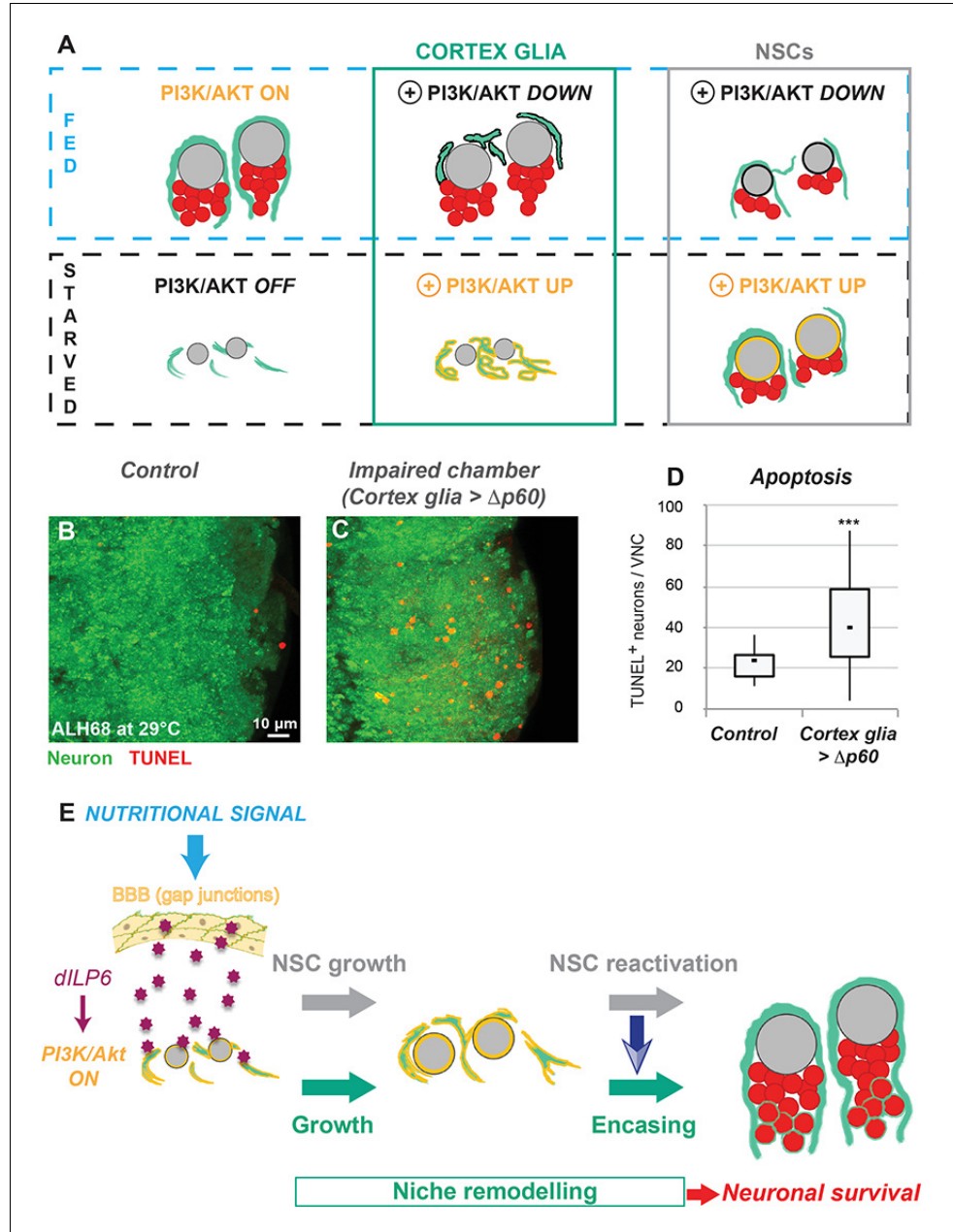

**Figure 5.** Nutrition-dependent formation of the cortex glial chamber promotes survival of newly-born neurons. (**A**) Selective manipulations of the PI3K/Akt pathway identify both autonomous and non-autonomous signals required for chamber formation. (**B–D**) Neuronal apoptosis is increased when chamber formation is impaired. TUNEL stainings in one VNC from B) control and C) Δp60 overexpression. Ventral view. Neuron, ElaV, green ; TUNEL, red. (**D**) Statistical analysis. p=2.9*10$^{-2}$. n (control) = 10 VNCs. n (Δp60)=9 VNCs. (**E**) Model of cortex glial chamber remodelling (see text for details).

DOI: https://doi.org/10.7554/eLife.30413.011

The following figure supplements are available for figure 5:

**Figure supplement 1.** Role and formation of the cortex glial chamber.
DOI: https://doi.org/10.7554/eLife.30413.012

**Figure supplement 2.** Impairing dynamin-dependent signalling in NSCs has limited impact on chamber formation.
DOI: https://doi.org/10.7554/eLife.30413.013

a small reduction in cortex glial membranes. This suggests that dynamin-dependent signalling in NSCs is not a major player required to complete chambers around each NSC lineage. Further investigation will be required to identify the main molecular cues from the NSCs, and how they are transmitted to the cortex glia.

How the cortex glial chamber protects newborn neurons from apoptosis remains to be understood. In the adult mammalian brain, astrocytes are key players supporting the different stages of neurogenesis (*Gengatharan et al., 2016*). This includes neuronal survival, by providing ion buffering and neurotransmitter recycling (*Sloan and Barres, 2014*). However, to what extent astrocytes adapt to environmental changes while maintaining these protective roles is as yet unknown. We show that early in *Drosophila* larval development, cortex glia architecture allows BBB-derived insulin-like peptides to reach NSCs during reactivation. Only once NSC proliferation has become nutrition-independent (*Cheng et al., 2011*), is cortex glial chamber formation fully achieved. This illustrates the importance of cortex glial integration of both environmental and NSC signals. In mammals, BBB endothelial cells also constitute a source of diffusible and contact-dependent signals important for NSC maintenance, proliferation and differentiation (*Goldman and Chen, 2011*; *Tavazoie et al., 2008*; *Ottone et al., 2014*). Interestingly, mammalian NSCs and endothelial cells closely associate at specific sites lacking pericyte and astrocyte covering (*Tavazoie et al., 2008*). Thus, a remarkable coordination between niche architecture and signalling across the BBB exists in both mammals and Drosophila. In the light of our findings, it would be of great interest to see if and how the BBB, astrocyte and NSC cross talk adapts to external and internal changes whilst supporting neurogenesis in the mammalian brain.

## Materials and Methods

### Genetics
The following stocks were used: *Nrv2::GFP* (protein trap FlyTrap, BDSC stock 6828), *NP2222-GAL4* (*Hayashi et al., 2002*; *Awasaki et al., 2008*), *cyp4g15-GAL4* (Janelia line, BDSC stock 50472, this study), *grh-GAL4* (*Chell and Brand, 2010*), *insc-GAL4* (*Mz1407* from J. Urban and G. Technau), *UAS-PTEN* (*Goberdhan et al., 1999*), *UAS-Δp60* (*Weinkove et al., 1999*), *UAS-myrAkt* (*Stocker et al., 2002*), *UAS-stg Shmir* (TRiP line, BDSC stock 34831), *UAS-dap* (*Lane et al., 1996*), *UAS-east$^{RNAi}$* (BDSC stock 33879), *UAS-shi$^{ts1}$* (BDSC stock 44222), *UAS-InR.K1409A; UAS-InR.K1409A* (BDSC stocks 8252 and 8253, referred as *2X InR$^{DN}$*).

All RNAi experiments were conducted at 29°C. The following induction times were used:

- *Figure 3A–F*, *Figure 4A–F* and *Figure 2—figure supplement 1A–A'*, *Figure 4—figure supplement 1* : from the start of GAL4 driver expression (late embyonic stages).
- *Figure 3G–L*, *Figure 4G–L* and *Figure 5B–D*, and *Figure 3—figure supplement 1*, *Figure 3—figure supplement 2* and *Figure 5—figure supplement 1D–G*: from larval hatching.

All cortex glia experiments were performed with *cyp4g15-GAL4* as driver, unless stated otherwise in the figure. NSC drivers were *insc-GAL4* (MZ1407) for *Figure 4A–F* and *Figure 4—figure supplement 1* and *Figure 5—figure supplement 2*; and *grh-GAL4* for *Figure 4G–L* and *Figure 1—figure supplement 1C–D''*.

### Larval culture
Larvae that hatched within a 60 min window (defined as after larval hatching 0 hr, or ALH0) were transferred to fresh yeast on a standard fly food plate. Starvation was achieved by transferring larvae to a solution of 20% sucrose in PBS. The developmental time is at 25°C, unless stated otherwise.

### Shibire experiments
We drove the thermosensitive allele of dynamin *shibire$^{ts1}$* with the *Nrv2::GFP,insc-GAL4/CyO; tubulin-GAL80$^{ts}$* driver line, in order to minimise its activity at permissive temperature (18°C) (*Gonzalez-Bellido et al., 2009*). The embryos were kept at 18°C during development and larvae were transferred to 31°C from ALH0. Larvae were dissected at ALH44.

## Immunohistochemistry

Larval brains were dissected according to standard procedures. Primary antisera were: Chicken anti-GFP (1/2000, 06–896, Upstate), Guinea Pig anti-Deadpan (*Caygill and Brand, 2017*) (1/5000), rat anti-ElaV (1/100, 7E8A10, DSHB), mouse anti-Repo (1/100, 8D12, DSHB), rabbit anti-PH3 (1/100, 06–570, Millipore). Samples were analysed with an Olympus upright or inverted FV1000 confocal microscope, or with a Zeiss LSM880 microscope.

## Image processing

Volocity or Fiji were used to process confocal data. Figures were assembled using Adobe Photoshop and Illustrator.

## Statistics

Experiments were not randomised or double blind. No statistical methods were used to determine sample size *a priori*. One biological replicate is defined as the result of one parental cross. For each experiment, at least two biological replicates were performed and analysed qualitatively for phenotype reproducibility. One biological replicate was quantified for generating chamber organisation charts and cortex glia/NSC ratios (*Figure 2E–F*; *Figure 3E–F and K–L*; *Figure 4E–F and K–L*; *Figure 3—figure supplement 1C–F*; *Figure 3—figure supplement 2H–L*, *Figure 4—figure supplement 1F–I* and *Figure 5—figure supplement 2C–F*). Two biological replicates were quantified for TUNEL staining (*Figure 5D*), with the exclusion of one outlier brain in the Δp60 dataset due to total absence of TUNEL +cells (including in the optic lobe, where apoptosis normally occurs). Two biological replicates were quantified for NSC number and cell cycle markers (*Figure 5—figure supplement 1A,B,D and E*). One biological replicate was quantified for glial cell number (*Figure 3—figure supplement 2D*), neuronal volume (*Figure 5—figure supplement 1C*) and mitotic index (*Figure 4—figure supplement 1C* and *Figure 5—figure supplement 2G*).

Bar graphs were generated using the mean and standard error of the mean (SEM) for each sample. Statistical significance was determined using an unpaired Student's test. *** represents p<0,05 (confidence interval of 95%), and * represents p<0,1 (confidence interval of 90%). Whisker plots were drawn using the minimum, quartile 1, median, quartile three and maximum of each condition's sample. In addition, a Student's test based on a sample's average and standard deviation was used to generate p values. *** represents p<0,05 (confidence interval of 95%), and * represents p<0,1 (confidence interval of 90%). Sample sizes and exact p values are indicated in the legend for each experiment. NSC and glia numbers were determined either using measurement scripts in Volocity software or manually.

## TUNEL staining

Staining was achieved using the ApopTag Red In situ apoptosis detection kit (Chemicon), following manufacturer's protocol. Modifications were as follows: brains were fixed for 20 min in 4% methanol-free formaldehyde and post-fixed in pre-cooled EtOH/PBS (2:1) for 5 min at −20°C. A 30 min incubation in10mM Sodium Citrate pH 6.0 at 70°C was performed before TdT staining to reduce background. TdT reaction was incubated for 3 hr at 37°C. Anti-dig rhodamine and primary antibodies were added together overnight at 4°C, and secondary detection was performed.

## EdU staining

EdU stainings were performed using Click-It Edu Alexa Fluor 594 (Thermo Fisher Scientific, MA, USA), following manufacturer's protocol. Brains were dissected inside-out in PBS and incubated for 45 min in EdU. They were washed once in PBS, then fixed for 20 min in 4% methanol-free formaldehyde before performing EdU staining.

## Chamber organisation chart

We developed a tailored Python script to analyse the changes in marker intensity against depth (Z-axis) for a confocal stack (F.N. Murphy). The tracked markers are cortex glial membrane (Nrv2::GFP, green), NSC nuclei (Dpn, grey) and neuronal nuclei (ElaV, red). The script allows a subregion of the sample to be selected by drawing a bounding box in the X,Y plane. This is necessary in the real

world where only a portion of the image may be relevant. For example, a brain image may contain substantial amounts of trachea (stained by Nrv2::GFP) that must be avoided.

The average intensity is computed within the bounded region for each layer of the stack and for each channel. This gives an average intensity versus depth curve for each channel. Since it is the distribution of marker that is of interest here, not its absolute value, the curves are normalised such that the peak intensity for each channel is one. A quadratic is fitted to the peak intensity sample and its two immediate neighbours for each channel. Differentiating the quadratic allows the position of the curve's peak to be estimated with precision better than the stack spacing. Each channel's curve is plotted with depth on the X-axis and averaged intensity on the Y-axis. A vertical line is plotted through the estimated peak position for each curve to facilitate reading the peak position against the X-axis scale.

### Cortex glial membrane measurements

For each VNC we selected a portion of the image, centred on thoracic NSCs, that is devoid of tracheal signal. We measured NSC numbers as well as the total intensity of the cortex glial membrane signal (Nrv2::GFP, green). For each experiment (control and experimental conditions), one specific threshold was chosen that would best fit the actual staining and kept throughout the analysis. The GFP intensity divided by NSC numbers represents the ratio of cortex glial membrane to NSCs.

### VNC neuronal volume measurements

For each VNC, we selected only the thoracic portion. We measured the total volume of ElaV staining using one specific threshold that was kept constant between all samples and conditions. We then divided each value by the average of control volume, in order to center control volume at 1.

### Transcriptional profiling

We used Targeted DamID (*Southall et al., 2013*; *Marshall et al., 2016*; *Marshall and Brand, 2015*) to profile transcription in subperineurial glial cells (Jessie van Buggenum, P.S. and A.H.B., unpublished). *UAS-Dam and UAS-Dam PolII* flies were crossed to *moody-GAL4; tubulin-GAL80^{ts}* flies. Their progeny were raised at 18°C then shifted to 29°C at ALH0 and dissected at ALH22. Processing of genomic DNA and data analysis were performed as previously described (*Southall et al., 2013*; *Marshall et al., 2016*; *Marshall and Brand, 2015*). Three biological replicates were performed. 3209 genes were called with a FDR < 0.01. Only *dilp6* showed significant binding of Dam-PolII (log2 ratio = 1.72 and FDR = 0.0029).

## Acknowledgements

We thank J van Buggenum for generously providing Supplementary Figure 2B Figure 2-figure supplement 1B and FN Murphy for help in developing the Python script. This work was funded by the Royal Society Darwin Trust Research Professorship, Wellcome Trust Senior Investigator Award 103792 and Wellcome Trust Programme grant 092545 to AHB and by an Institut Pasteur/LabEx starting grant to PS. AHB acknowledges core funding to the Gurdon Institute from the Wellcome Trust (092096) and CRUK (C6946/A14492).

## Additional information

### Funding

| Funder | Grant reference number | Author |
|---|---|---|
| Institut Pasteur | ANR-10-LBX-73 S-CR11019-34 | Pauline Spéder |
| Wellcome | Senior Investigator Award 103792 | Andrea Brand |
| Wellcome | Programme Grant 092545 | Andrea H Brand |
| Wellcome | 092096 | Andrea Brand |
| Cancer Research UK | C6946/A14492 | Andrea Brand |

| Royal Society | Royal Society Darwin Trust Research Professorship | Andrea H Brand |

The funders had no role in study design, data collection and interpretation, or the decision to submit the work for publication.

## Author contributions
Pauline Spéder, Conceptualization, Resources, Formal analysis, Investigation, Methodology, Writing—original draft, Project administration, Writing—review and editing; Andrea H Brand, Conceptualization, Resources, Data curation, Formal analysis, Supervision, Funding acquisition, Investigation, Methodology, Writing—original draft, Project administration, Writing—review and editing

## Author ORCIDs
Pauline Spéder (iD) https://orcid.org/0000-0002-2934-5841
Andrea H Brand (iD) https://orcid.org/0000-0002-2089-6954

## Decision letter and Author response
Decision letter https://doi.org/10.7554/eLife.30413.016
Author response https://doi.org/10.7554/eLife.30413.017

## Additional files

### Supplementary files
• Transparent reporting form
DOI: https://doi.org/10.7554/eLife.30413.014

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
