## [Decision Letter]

The reviews for your manuscript are all generally very positive, and the reviewers and I agree that the manuscript is suitable for *eLife* subject to a number of specific changes that are suggested. Although the list of suggestions is long, it is my feeling that they can be addressed rather easily either by rewriting the text or by a few simple experiments. Before sending a formal acceptance, I wanted to confirm that you will be able to make these. Please send your responses so that we can consult with the reviewers if necessary and then move to the next step. I am attaching the verbatim reviews.

Reviewer #1:

In this study, Brand and Spéder identify that systemic nutritional as well as NSC-derived local signals modify the cortex glia, which in turn promotes survival of new-born neurons. Cortex glia evolve throughout larval development by undergoing 3 distinctive stages: membrane expansion, formation of NSC chamber and glial extension, which is closely co-related to reactivation and development of NSC. Authors have showed that starvation prevents proper formation of the cortex glial chamber, of which phenotype is recapitulated either by disrupting BBB or by expression of PI3K_DN in the cortex glia. However, constitutive activation of AKT in the cortex glia upon starvation is not sufficient to restore neither NSC nor the cortex glia organization. These results indicate that autonomous PI3K/AKT activity in the cortex glia is not enough to establish its own structure. Interestingly, blocking NSC reactivation strongly affects the cortex glia organization and forced NSC reactivation upon starvation is sufficient to restore the cortex glia, suggesting that NSC reactivation is necessary and sufficient for the cortex glia formation upon starvation. Finally, authors also have showed that cortex glia formation is particularly important for the survival of new-born neurons as lack of cortex glia induces neuronal cell death.

This work is straightforward and interesting in that it suggests dual modifications of the NSC niche development in association with the NSC development. However, I found the study requires further supports for mechanistic details underlying these phenomena, which are very central points for the paper.

1) Although previous studies have identified that Dilp6 originated from the surface glia is essential for the NSC reactivation via BBB, it is unclear whether cortex glia development is also controlled by the same insulin. Given that this study highlights dual (systemic and local) controls of the cortex glia, it is important to show the specific Dilp involved in this control.

2) Authors claimed that membrane expansion of the cortex glia is mainly regulated by systemic factors while organization is separable and controlled by NSC-derived local signal. If NSC-derived local signal only contributes to the organization of the cortex glia, timely control of NSC activity during development will differentially influence the cortex glial structure. Current version of study lacks detail developmental analysis on distinctive control of expansion or organization of the cortex glia, which would further strengthen the relevance of findings.

3) Correlated to point 1 and 2; Figure 4, expression of Akt_act in the NSC upon starvation at 96 ALH shows almost complete restoration of cortex glia formation compare to fed control. Moreover, impaired PI3K/AKT in the NSC is sufficient to hinder the cortex glia formation, suggesting that local signal from NSC is dominant over the systemic signal in cortex glia development. Or it is possible that systemic information is mainly delivered to the NSC that in turn activate the cortex glia development. Figure 3' is the only experiment that shows the systemic contribution; therefore, it requires additional analysis including modulation of InR directly to support the systemic control.

4) Figure 5 can be improved by showing cortex glia structure, neuron and TUNEL+ cells in detail to highlight that it is impaired cortex glia that alters survival of new-born neuron.

5) Supplemental Figure 5 shows the chamber disruption phenotype of Trol_null mutant to indicate that Trol could be a local activator of glial chamber formation originated from the NSC. However, to claim this, it requires further extensive analysis including expression profile of Trol, NSC-specific knock-down of Trol and rescue, and so on.

Reviewer #2:

This paper investigated the interactions between cells in the neuronal stem cell (NSC) niche of the ventral nerve cord (VNC) in larval *Drosophila*, and also the role of systemic inputs to this niche. Neuronal stem cells when re-activated in the larva start proliferation and give rise to a progeny of neurons. During this process niche cells are enclosed in a microenvironment consisting of NCS, cortex glia and neurons. The encasement formed by cortex glia is vital for proper differentiation and survival of the newborn neurons and the cellular interactions giving rise to this tailored niche was not known in detail prior to the present study. The authors first describe the sequence of morphological events in the formation of the encasement, then move on to show how nutritional signals, cells of the blood brain barrier (BBB), NSCs and cortex glia interact in a sequence to ensure proper differentiation of the niche cells. Signals include nutrients, insulin-IGF signaling (IIS) and some unknown factor (matrix protein?). The study takes us a small step forwards in understanding how the niche develops and its importance for survival of newborn neurons, but does not impress me as a huge conceptual leap. However, the paper seems to be written as a short communication, and as such it may be alright that the story is not complete. The schematic figures (in Figure 1, Figure 5—figure supplement 1) are really instructive.

I have some rather minor queries and suggestions for improvement of the paper.

1) The paper is apparently some form of short communication and the authors have only one header: Introduction and Results. What about Discussion/Conclusions? Usually Results and Discussion are merged (with a separate Introduction) or there are no headers at all (e. g. as in Nature letters).

2. In the Introduction: the authors say "… a simpler model, the postembryonic larval brain of…". However, they study the simpler VNC, not the brain (which is far more complex, and probably not equally suitable for this kind of study).

3. In the Introduction the authors mention that the BBB mediates the impact of nutrition on the NCS reactivation and that one BBB derived signal is insulin like peptides (DILPs) that bind to the receptor (InR) on NCSs. However, the authors do not really define what constitutes the BBB. What cells are forming this tissue and which of them produce DILPs? Presumably the same cells are nutrient sensing. Is it known what the sensor is? Is it slimfast or some similar transporter? I ask this because in mammals the BBB is formed by several cell types and structures. Only in the Supplemental Figure 1 there is an indication that the BBB is composed of specific cells (glial cells) – but no mention in the text. Would be useful to have some more detailed information on the BBB in *Drosophila* in the Introduction.

4. A minor detail: the authors use dilp and dilps for the peptide(s) – usually the peptide is written DILP (DILPs) or dILP (in contrast to the gene or transcript, dilp).

5. In the Introduction: "BBB secreted insulin-like peptides" is it known which peptides? DILP6? DILP2? Other?

6. The text referring to Figure 4 is a bit confusing. The authors say that they do not show deltap60 data, but they do in Figure 4, conversely they say they show PTEN expression in the Figure 4', but they do not. Seems like the two were mixed here?

7. The experiment to show the role of NCS mitotic entry, rather than enlargement (growth), promotes chamber completion, by testing Perlecan mutants seems rather weak and preliminary as evidence.

Reviewer #3:

In their manuscript, Spéder and Brand address a largely unconsidered but important question of how cells that constitute stem cell niches (in this case cortex glia) adapt and remodel over time to meet the needs of the stem cells and stem cell progeny they harbor.

The authors focus on neural stem cells and neurons in the *Drosophila* ventral nerve cord. They are enveloped in discreet chambers by cortex glia. These glia respond to systemic nutritional status to reactivate quiescent neural stem cells through the secretion of insulin-like peptides. They characterize the changes in cortex glial membrane remodeling in relation to the quiescence and reactivation of the neural stem cells. They find that cortex glia undergo striking membrane remodeling in parallel with stem cell reactivation to form chambers around them and their neuronal progeny. The authors go on to show that PI3K activity in cortex glia leads to membrane expansion but not encapsulation or chamber formation. Instead they show that stem cell reactivation (by PI3K activity) is required for encapsulation. Finally, the authors show that while glial chamber formation or encapsulation is dispensable for stem cell proliferation and survival, it is required for the survival of newly formed neurons. Overall, this work nicely addresses an interesting aspect of niche cell dynamics in relation to changing stem cell needs and shows that two-way communication between niche cells and stem cells and/or their progeny is required to coordinate the process.

Nonetheless, the manuscript would be improved by addressing the following points:

1) The authors clearly showed that stem cell reactivation is required for chamber formation. A major focus of the manuscript is the argument that NSCglial signals promote chamber formation. However, the experiments in the manuscript cannot distinguish whether the relevant encapsulation-promoting signals arise from the stem cells (as the authors assert) or from the neuronal progeny born when these cells become reactivated. This is an important point to clarify as it is strongly emphasized in the text. For example, how does encapsulation proceed when neuronal progeny are killed off? How does Shibirets expression in stem cells or neurons affect chamber formation?

2) If the relevant signals do indeed arise from stem cells, they could be either mechanical, or secreted as suggested by the authors. They note chamber formation defects in perlecan mutants and suggest that "NSC mitotic entry, rather than enlargement, promotes chamber completion around each NSC lineage". The link between defects in perlecan mutants and stem cell mitotic entry is not clear. What is the source of perlecan (systemic or local)? Whole animal perlecan mutants are very disrupted. How does overexpression of matrix metalloproteinases in either stem cells or neurons affect chamber formation?

3)The authors show that glial chamber formation is dispensable for stem cell proliferation and survival, thus in this context once reactivated by glial insulins, stem cells carry on independently of the niche. On the other hand neuronal progeny survival depends on glial chamber formation. Thus, rather than providing a niche for supporting stem cells, cortex glia may be providing a differentiation niche for neurons. Do these neurons differentiate appropriately if neuronal death is prevented by overexpression of P35 when chamber formation is impaired?

---

## [Author Response]

Reviewer #1:1) Although previous studies have identified that Dilp6 originated from the surface glia is essential for the NSC reactivation via BBB, it is unclear whether cortex glia development is also controlled by the same insulin. Given that this study highlights dual (systemic and local) controls of the cortex glia, it is important to show the specific Dilp involved in this control.

We previously profiled gene expression in subperineurial glia from ALH0 to ALH22 at 29°C (thus encompassing most of the expansion phase of the cortex glia) using our Targeted DamID technique (Southall et al., 2013). We found that only dilp6 was expressed. We have added the relevant data to Figure 2—figure supplement 1 and modified the text in the Results section to read:

“To identify which dILPs are expressed by the BBB at the time of membrane growth, we determined RNA Pol II binding in the subperineurial glia of fed larvae (Jessie Van Buggenum, P.S. and A.H.B., unpublished). We found Pol II binding only at dilp6 (Figure 2—figure supplement 1).”

2) Authors claimed that membrane expansion of the cortex glia is mainly regulated by systemic factors while organization is separable and controlled by NSC-derived local signal. If NSC-derived local signal only contributes to the organization of the cortex glia, timely control of NSC activity during development will differentially influence the cortex glial structure. Current version of study lacks detail developmental analysis on distinctive control of expansion or organization of the cortex glia, which would further strengthen the relevance of findings.

Whereas cortex glial membrane growth can be influenced by internal or external inputs, we propose that cortex glial organisation is mostly, if not only, directed by signals from NSC lineages. Manipulation of the PI3K/Akt pathway in the cortex glia shows that it is necessary for membrane growth but not sufficient for chamber organisation. This suggests that membrane growth does not lead per se to NSC encapsulation and that the two elements are indeed separable. It also suggests that an additional input is needed to convert this growth into a tailored architecture, anchoring it to NSC lineages. In addition, our analysis of the developmental timing of chamber formation (Figure 1) shows that substantial membrane growth occurs before complete NSC reactivation. Delaying NSC reactivation affects glial architecture more than membrane growth, whilst forcing NSC reactivation is able to lead to NSC encapsulation (Figure 4). This shows that reactivated NSCs are crucial to direct cortex glial architecture.

To strengthen the idea that impairing NSC reactivation mainly affects cortex glial organization, we added the following text to the Results section:

“To confirm the requirement for NSC reactivation in cortex glial chamber organisation, we turned to intrinsic regulators of this process. Spindle matrix proteins have recently been shown to be required for the mitotic re-entry of quiescent NSCs^27^. Accordingly, RNAi knockdown of east in NSCs resulted in a dramatic loss of the mitotic marker phosphohistone H3 (PH3, Figure 4—figure supplement 1). We found that this condition also led to impaired cortex glial chamber organisation (Figure 4—figure supplement 1), whereas membrane growth was little affected (Figure 4—figure supplement 1). These results show that NSC reactivation is crucial to chamber organisation, but is not the main driving force for cortex glial membrane growth under normal, fed conditions.”

3) Correlated to point 1 and 2; Figure 4, expression of Akt_act in the NSC upon starvation at 96 ALH shows almost complete restoration of cortex glia formation compare to fed control. Moreover, impaired PI3K/AKT in the NSC is sufficient to hinder the cortex glia formation, suggesting that local signal from NSC is dominant over the systemic signal in cortex glia development. Or it is possible that systemic information is mainly delivered to the NSC that in turn activate the cortex glia development. Figure 3' is the only experiment that shows the systemic contribution; therefore, it requires additional analysis including modulation of InR directly to support the systemic control.

To show that direct activation of PI3K/AKT through dILP binding to InR is required for proper chamber formation, we expressed two copies of a dominant negative form of the insulin receptor (2X InR DN) in the cortex glia. This heavily impaired chamber formation. These data have been added as Figure 3—figure supplement 1 and the text in the Results section now reads:

“Notably, affecting insulin signalling directly at the level of the insulin receptor, by expressing a dominant negative form (2X InR^DN^), also led to impaired chamber formation, with loss of cellular organisation and strongly decreased membrane signal (Figure 3—figure supplement 1). These data demonstrate that direct integration of dILP6 binding and insulin signalling are required in the cortex glia for NSC chamber building.”

4) Figure 5 can be improved by showing cortex glia structure, neuron and TUNEL+ cells in detail to highlight that it is impaired cortex glia that alters survival of new-born neuron.

The extent of cortex glial disruption after Δp60 expression is too severe to be able to correlate localised chamber defects with increased TUNEL staining in the ventral nerve cord.

5) Supplemental Figure 5 shows the chamber disruption phenotype of Trol_null mutant to indicate that Trol could be a local activator of glial chamber formation originated from the NSC. However, to claim this, it requires further extensive analysis including expression profile of Trol, NSC-specific knock-down of Trol and rescue, and so on.

These data were mainly intended to support discussion, and we agree that they are too preliminary. We have removed them.

Reviewer #2:1) The paper is apparently some form of short communication and the authors have only one header: Introduction and Results. What about Discussion/Conclusions? Usually Results and Discussion are merged (with a separate Introduction) or there are no headers at all (e. g. as in Nature letters).

We have now three headings: Introduction, Results and Discussion sections.

2. In the Introduction: the authors say "… a simpler model, the postembryonic larval brain of…". However, they study the simpler VNC, not the brain (which is far more complex, and probably not equally suitable for this kind of study).

We have changed brain for central nervous system (CNS) or ventral nerve cord (VNC) throughout the text.

3. In the Introduction the authors mention that the BBB mediates the impact of nutrition on the NCS reactivation and that one BBB derived signal is insulin like peptides (DILPs) that bind to the receptor (InR) on NCSs. However, the authors do not really define what constitutes the BBB. What cells are forming this tissue and which of them produce DILPs? Presumably the same cells are nutrient sensing. Is it known what the sensor is? Is it slimfast or some similar transporter? I ask this because in mammals the BBB is formed by several cell types and structures. Only in the Supplemental Figure 1 there is an indication that the BBB is composed of specific cells (glial cells) – but no mention in the text. Would be useful to have some more detailed information on the BBB in Drosophila in the Introduction.

The Results section now reads:

“The *Drosophila* BBB acts as a signalling interface between the hemolymph (the *Drosophila* equivalent of blood) and brain cells^11,12^. It is exclusively of glial nature, formed by a layer of perineurial glia and a layer of subperineurial glia, while the vertebrate BBB is a composite of endothelial cells and glial cells (astrocytes) ^11^. Both fulfill neuroprotective roles, relying on a physical paracellular barrier (tight junctions of the vertebrate endothelial cells and septate junctions of the *Drosophila* subperineurial glia). Importantly, the *Drosophila* BBB mediates the impact of nutrition on NSC reactivation^13,14^. Essential amino acids in the larval diet trigger the local production and secretion of insulin-like peptides (dILPs) by the subperineurial glial layer of the BBB^13,14^.”

4. A minor detail: the authors use dilp and dilps for the peptide(s) – usually the peptide is written DILP (DILPs) or dILP (in contrast to the gene or transcript, dilp).

We have now spelled the peptide “dILP”.

5. In the Introduction: "BBB secreted insulin-like peptides" is it known which peptides? DILP6? DILP2? Other?

We have now added data showing that only dilp6 is expressed in the BBB (Figure 2—figure supplement 1).

6. The text referring to Figure 4 is a bit confusing. The authors say that they do not show deltap60 data, but they do in Figure 4, conversely they say they show PTEN expression in the Figure 4', but they do not. Seems like the two were mixed here?

We have rectified this.

7. The experiment to show the role of NCS mitotic entry, rather than enlargement (growth), promotes chamber completion, by testing Perlecan mutants seems rather weak and preliminary as evidence.

These data were mainly intended to support discussion and we agree that they are too preliminary. We have removed them.

Reviewer #3:1) The authors clearly showed that stem cell reactivation is required for chamber formation. A major focus of the manuscript is the argument that NSCglial signals promote chamber formation. However, the experiments in the manuscript cannot distinguish whether the relevant encapsulation-promoting signals arise from the stem cells (as the authors assert) or from the neuronal progeny born when these cells become reactivated. This is an important point to clarify as it is strongly emphasized in the text. For example, how does encapsulation proceed when neuronal progeny are killed off? How does Shibirets expression in stem cells or neurons affect chamber formation?

We agree that the impact of NSC reactivation on chamber formation could be indirect, and the result of producing progeny that will subsequently signal to the cortex glia. Our main reason to propose NSCs as the major signalling cells is the timing of encapsulation with respect to neuronal production. NSCs are encapsulated between ALH24 and ALH48, prior to any substantial neuronal production. Furthermore, forcing PI3K/AKT activation in starved NSCs leads to their encapsulation whereas neuronal production is minimal, as the larvae do not survive.

The text in the Results section now reads:

“Importantly, our work reveals a previously unidentified NSC lineage to cortex glial signal. NSC reactivation, sufficient for triggering lineage encasing, leads to the production of new neurons. NSC lineages are enclosed at about the time of the first NSC division, before any substantial production of neurons. Although we cannot completely rule out that neurons contribute to lineage encasing, we propose that NSCs are the signal-sending cells due to the timing of the event.

The instructive signals sent by NSCs to remodel the cortex glia could be either mechanical or secreted. Interestingly, blocking endocytosis and vesicle recycling in NSCs through dynamin inhibition (Figure 5—figure supplement 2) has a limited impact on chamber formation, mostly leading to a small reduction in cortex glial membranes. This suggests that dynamin-dependent signalling in NSCs is not a major player required to complete chambers around each NSC lineage. Further investigation will be required to identify the main molecular cues from the NSCs and how they are transmitted to the cortex glia.”

2) If the relevant signals do indeed arise from stem cells, they could be either mechanical, or secreted as suggested by the authors. They note chamber formation defects in perlecan mutants and suggest that "NSC mitotic entry, rather than enlargement, promotes chamber completion around each NSC lineage". The link between defects in perlecan mutants and stem cell mitotic entry is not clear. What is the source of perlecan (systemic or local)? Whole animal perlecan mutants are very disrupted. How does overexpression of matrix metalloproteinases in either stem cells or neurons affect chamber formation?

The Perlecan data were mainly intended to support discussion and we agree that they are too preliminary. We have removed them.

3)The authors show that glial chamber formation is dispensable for stem cell proliferation and survival, thus in this context once reactivated by glial insulins, stem cells carry on independently of the niche. On the other hand neuronal progeny survival depends on glial chamber formation. Thus, rather than providing a niche for supporting stem cells, cortex glia may be providing a differentiation niche for neurons. Do these neurons differentiate appropriately if neuronal death is prevented by overexpression of P35 when chamber formation is impaired?

We agree that this is an interesting point but believe it is beyond the scope of this paper. It would require setting up two different binary systems in order to impair chamber formation while driving p35 in the neurons.